# *Bcl-2* Orthologues, *Buffy* and *Debcl*, Can Suppress *Drp1*-Dependent Age-Related Phenotypes in Drosophila

**DOI:** 10.3390/biom14091089

**Published:** 2024-08-30

**Authors:** Azra Hasan, Brian E. Staveley

**Affiliations:** Department of Biology, Memorial University of Newfoundland, St. John’s, NL A1C 5S7, Canada; ahasan@mun.ca

**Keywords:** *Drp1*, *Buffy*, *Debcl*, *Drosophila melanogaster*, Amyotrophic Lateral Sclerosis, Parkinson’s disease, mitochondrial fission, mitochondrial dysfunction

## Abstract

The relationship of Amyotrophic Lateral Sclerosis, Parkinson’s disease, and other age-related neurodegenerative diseases with mitochondrial dysfunction has led to our study of the mitochondrial fission gene *Drp1* in *Drosophila melanogaster* and aspects of aging. Previously, the Drp1 protein has been demonstrated to interact with the Drosophila Bcl-2 mitochondrial proteins, and *Drp1* mutations can lead to mitochondrial dysfunction and neuronal loss. In this study, the *Dopa decarboxylase-Gal4* (*Ddc-Gal4*) transgene was exploited to direct the expression of *Drp1* and *Drp1-RNAi* transgenes in select neurons. Here, the knockdown of *Drp1* seems to compromise locomotor function throughout life but does not alter longevity. The co-expression of *Buffy* suppresses the poor climbing induced by the knockdown of the *Drp1* function. The consequences of *Drp1* overexpression, which specifically reduced median lifespan and diminished climbing abilities over time, can be suppressed through the directed co-overexpression of pro-survival *Bcl-2* gene *Buffy* or by the co-knockdown of the pro-cell death *Bcl-2* homologue *Debcl*. Alteration of the expression of *Drp1* acts to phenocopy neurodegenerative disease phenotypes in Drosophila, while overexpression of *Buffy* can counteract or rescue these phenotypes to improve overall health. The diminished healthy aging due to either the overexpression of *Drp1* or the RNA interference of *Drp1* has produced novel Drosophila models for investigating mechanisms underlying neurodegenerative disease.

## 1. Introduction

The mitochondrial network is essential for many aspects of the subcellular survival mechanisms of organisms. Known to be the “powerhouse of the cell”, mitochondria are responsible for various aspects of energy homeostasis, oxidative stress, calcium handling, cell signalling, and, thus, cell survival [1,2,3]. The dynamic nature of the mitochondria population is critical to the integrity of the subcellular network structures that these organelles maintain and to the control of the quality of mitochondrial proteins and other components [4,5]. For example, the early dynamic events that occur during apoptosis include cristae remodelling, mitochondrial fragmentation, and membrane “blebbing”. Inhibition of these processes, either through knockdown directed by RNA interference (RNAi) or by expression of a dominant-negative mutant form of *dynamin-related protein 1 (Drp1)*, can slow the rate of mitochondrial fragmentation and, in turn, the cascade of apoptotic events. The activity of the Drp1 protein promotes caspase-independent mitochondrial fission and cristae remodelling to amplify the process of apoptosis, whether or not cell death is instigated by either the specific activity of the pro-apoptotic protein BH3 interacting-domain death agonist (BID) or by the general consequences of oxidative stress [6]. Overall, the mechanics of mitochondrial fission plays a crucial role in the amplification of aspects of the essential cellular process of apoptosis.

Modifications to the mitochondrial network seem to differentially influence a number of signalling pathways. In response to a series of molecular cues [7,8], the mitochondrial network participates in a delicate balance between the continuous division and fusion processes [9]. Mitochondrial fusion helps compromised mitochondria, likely bearing highly damaged DNA and proteins, to actively exchange components with other more healthy mitochondria. This process acts to decrease the severity of this accumulated heteroplasmy (differing mitochondrial sub-lineages) and help with functional complementation [10]. Recently, much has been understood about mitochondrial fission, a fundamental process in the maintenance of the mitochondrial network, and the requirements for *Drp1* [2,3]. Yet, a complete understanding of the factors that control mitochondrial dynamics during age-related processes [11] remains limited.

The B-cell lymphoma-2 (Bcl-2) family of proteins interact with the Drp1 protein, and expression of *Drp1* can promote apoptosis in both Bcl-2 protein-dependent and independent manners [6]. The two Bcl-2 family homologues in *Drosophila melanogaster* are *Buffy* (anti-apoptotic) and *Debcl* (pro-apoptotic) [12]. The Debcl protein can interact with Drp1 in Drosophila to activate apoptosis via the c-Jun N-terminal kinase (JNK) pathway [13]. *Drp1* is required for a standard rate of Cyt-c release and caspase activation during programmed cell death. The Drp1 protein interacts with other proteins involved in a number of mitochondrial processes, such as the protein product of *Bax* [8]. Diverse stresses can increase the translocation of cytosolic Drp1 to the mitochondria, thus leading to the induction of excessive fragmentation and initiation of apoptosis or mitophagy [14]. In a number of models of ALS, dephosphorylation of Drp1 through the activity of protein phosphatase 1 has been identified as causative of the disease-associated phenotypes [15]. As well, Drp1-mediated mitochondrial fragmentation caused by the administration of rotenone has been identified in a rat model of PD-like changes to the olfactory bulb [16]. As adjustment of *Drp1* gene activity may be key to the inhibition of the pathology of ALS and PD, investigation of such alterations may provide some knowledge helpful in the development of therapies. Excessive activity of Drp1 increases mitochondrial fission and consequently promotes cell death and/or degeneration.

Here, we show that alteration of *Drp1* expression can result in distinct phenotypes over time. Our research group employs “the fruit fly”, *Drosophila melanogaster*, to model neurodegenerative disease because it is an excellent model system for studying the genes and proteins affected in ALS, PD, and aging [17]. The anticipated role of mitochondria in pathogenesis has made the study of the interactions of the *Drp1* gene important for modelling these diseases in Drosophila. In these experiments, we exploited the *UAS-Gal4* system to direct the overexpression and knockdown of the genes of interest in selected neuronal tissues, using the *Ddc-Gal4* transgene [18]. We propose that the *Drp1* overexpression phenotype is due to excessive activities related to apoptosis and can be rescued by the appropriate regulation by the anti-apoptotic *Bcl-2* gene, *Buffy*. The knockdown phenotype produced in response to the expression of *Drp1-RNAi* may be due to the diminishment of mitochondrial integrity and may be rescued through modification of the responsible signalling pathway. The careful regulation of mitochondrial dynamics must be important in the control of age-related mitochondrial-induced defects. Overall, our strategy is to identify the basic mechanisms in the fly model to encourage further validation in mammalian model organisms.

## 2. Materials and Methods

### 2.1. Bioinformatic Analysis

Protein sequences were obtained from the National Center of Biotechnology Information (NCBI) database (https://www.ncbi.nlm.nih.gov/protein/, accessed on 1 July 2024). The conserved domains were identified using NCBI Conserved Domain Database (CDD) (https://www.ncbi.nlm.nih.gov/cdd/, accessed on 1 July 2024) and Eukaryotic Linear Motif (ELM) (http://elm.eu.org/, accessed on 1 July 2024). Multiple sequence alignment was performed using Clustal Omega (https://www.ebi.ac.uk/Tools/msa/clustalo/, accessed on 1 July 2024) to reveal the conservation of domains. The *Homo sapiens* Dynamin-1-like protein (DLP-1/Drp1) structure (PDB ID 4BEJ) was obtained from NCBI structure database (https://www.ncbi.nlm.nih.gov/structure/, accessed on 1 July 2024), and *Drosophila melanogaster Drp1* protein structure was developed using Phyre2 (http://www.sbg.bio.ic.ac.uk/phyre2/html/page.cgi?id=index, accessed on 1 July 2024) modelling tool. The final models were edited with the PyMOL version 2 software (https://pymol.org/2/, accessed on 1 July 2024) to highlight the N-terminus, C-terminus and consensus LC3-interacting region (LIR) regions.

### 2.2. Drosophila Stocks and Media

The stocks, *UAS-lacZ^4-1-2^*; (BDSC_1776: w; P{UAS-lacZ.B}Bg4-1-2); *UAS-Drp1* (BDSC_51647: y w; P{UAS-Drp1.D}3); *UAS-Drp1-RNAi^JF02762^* or *UAS-Drp1-RNAi1* (BDSC_27682: y v; P{TRiP.JF02762}attP2); *UAS-Drp1-RNAi^HMC03230^* or *UAS-Drp1-RNAi2* (BDSC_51483: y v; P{TRiP.HMC03230}attP40); *UAS-Buffy* (BDSC_58358: w; P{UAS-Buffy.Q}2); *UAS-Buffy-RNAi* (BDSC_32060: w; P{UAS-Buffy.RNAi}3); *UAS-Debcl^EY05743^* or *UAS-Debcl* (BDSC_20156: y w; P{EPgy}Debcl[EY05743]); and *Ddc-Gal4^4.3D^* (BDSC_7010: w; P{Ddc-Gal4}4.3D) were obtained from Bloomington Drosophila Stock Center (BDSC) at Indiana University, Bloomington, Indiana, USA. The *UAS-Debcl-RNAi^v47515^* (VDRC_47515: P{GD1637}v47515) was obtained from Vienna Drosophila Resource Center (VDRC). The *Ddc-Gal4/CyO*; *UAS-Drp1/TM3*, *Ddc-Gal4/CyO*; *UAS-Drp1-RNAi/TM3* derivative lines were generated using standard recombination methods and used to overexpress or inhibit *Drp1* in the selected DA neurons using the *Ddc-Gal4^4.3D^* transgene. In brief, a standard “double balancer chromosome” line of the genotype *w^1118^*; *L/CyO*; *Ki/TM3* was used to generate intermediate lines *w^1118^*; *Ddc-Gal4/CyO*; *Ki/TM3*; *w^1118^*; *L/CyO*; *UAS-Drp1/TM3* and *w^1118^*; *L/CyO*; *UAS-Drp1-RNAi/TM3* which were then used to generate the *w^1118^*; *Ddc-Gal4/CyO*; *Ki/TM3*; *UAS-Drp1/TM3* and *w^1118^*; *Ddc-Gal4/CyO*; *UAS-Drp1-RNAi/TM3* lines.

Sources of validation of the transgenes from previous studies that are used in this study are as follows: *Ddc-Gal4^4.3D^* (Gal4 expression validated in [18]); *UAS-lacZ^4-1-2^* (directed expression verified in [19]); *UAS-Drp1* (directed expression validated in [20]); *UAS-Drp1-RNAi(1)* (RNA interference validated in [21]); *UAS-Drp1-RNAi2* (RNA interference validated in [22]); *UAS-Buffy* (directed expression validated in [12]); *UAS-Buffy-RNAi* (RNA interference validated in [23]); *UAS-Debcl^EY05743^* (directed expression validated in [24]); and *UAS-Debcl-RNAi^v47515^* (RNA interference validated in [25]).

All stocks and experiments were maintained on a standard cornmeal/molasses/yeast/agar medium treated with propionic acid and methylparaben to resist fungal growth. Aliquots of media were poured into plastic vials, allowed to solidify, and refrigerated at 4 °C until used. Stocks were kept at room temperature while crosses and experiments were carried out at 25 °C. In all experimental runs, control critical class individuals were generated and evaluated in parallel under conditions nearly identical to the experimental flies.

### 2.3. Aging Assay

Crosses of select virgin females and males were made, and a cohort of critical class males was collected upon eclosion. At least 250 flies were aged per genotype in the cohorts of 25 or less per vial on fresh media, replenished every two to five days to avoid crowding. Flies were observed and scored every two days for the presence of deceased adults. As a rule, flies were considered dead when they did not display movement upon agitation [26]. Longevity data were analyzed using GraphPad Prism version 8 statistical software (graphpad.com), and survival curves were compared by the Mantel–Cox test. Significance was determined at a 95% confidence level (*p* ≤ 0.05) with Bonferroni correction.

### 2.4. Climbing Assay

A cohort of 70 critical-class male flies was collected within 24 h and maintained as a maximum of ten flies per vial at low density. The food was changed twice every week. Initially, every week, 50 males of each genotype were assayed, in groups of 10 or less, for their ability to climb a glass tube divided into five levels of 2 cm each according to standard protocol [26]. However, as flies died, the climbing assay was conducted on the survivors. The climbing index was calculated for each week using GraphPad prism version 8 statistical software. The climbing curve was fitted using non-linear regression and determined at a 95% confidence interval (*p* ≤ 0.05).

## 3. Results

### 3.1. Drp1 Is Highly Conserved between Homo sapiens and Drosophila melanogaster

The *D. melanogaster* Drp1 protein sequence was sourced from NCBI protein, and the conserved sequences were identified using NCBI CDD. NCBI protein Blast of Drp1 protein of *D. melanogaster* (NP_608694.2) with the *H. sapiens,* identified dynamin-1-like protein (isoform 4) (NP_001265392.1), it is 65% identical with a bit score of 957. The multiple sequence alignment of the two proteins derived by Clustal Omega (Figure 1a) shows a highly conserved dynamin-like protein family domain, a dynamin central domain, and a dynamin GTPase effector domain. Two well-documented phosphorylation sites are identified: S606 and S627 in *dynamin-1-like protein* isoform 4 of *H. sapiens*; and S616 and T637 in *Drp1* of *D. melanogaster*. A template-based modelling of *D. melanogaster* Drp1 protein by use of a combination of empirically derived energy functions and physics-based simulated folding was produced using Phyre2. The modelled *D. melanogaster* Drp1 protein (i) and the *H. sapiens* Dynamin-1-like protein (ii) from the NCBI database share a near identical structure (Figure 1b). The amino-terminus region of the Drp1 protein is highly conserved and has a consensus LC3-interacting region (LIR) sequence for binding to the ATG8/LC3 protein as determined by the Eukaryotic Linear Motif (ELM) resource. As this protein structure is so highly conserved, it seems very likely that the functions are highly conserved.

### 3.2. The Directed Overexpression and Knockdown of Drp1 with Ddc-Gal4^4.3D^

In this set of experiments, the control *Ddc-Gal4^4.3D^*; *UAS-lacZ* critical class males were determined to have a median lifespan of 68 days (n = 340). The directed overexpression of *Drp1* by the *Ddc-Gal4* transgene results in a decreased lifespan of 56 days in 314 flies, much lower compared to the control as determined by log-rank (Mantel–Cox) test with a *p*-value at <0.0001 (Figure 2A). Inhibition of *Drp1* by two distinct RNAi transgenes via the *UAS-Drp1-RNAi1* and *UAS-Drp1-RNAi2* directed by the *Ddc-Gal4* transgene results in median lifespans of 70 (n = 377) and 72 days (n = 323), respectively; very similar to the control (Figure 2A) as determined by log-rank (Mantel–Cox) test with *p*-value 0.0566 and 0.0213. The non-linear fitting of the climbing curve shows that altering the *Drp1* expression in either direction compromises the climbing ability phenotype compared to control at 95% confidence interval (CI) (*p*-value < 0.0001) (Figure 2B) (n = 50).

### 3.3. Phenotypic Rescue by Co-Expression of Drp1 and Drp1-RNAi Directed by Ddc-Gal4^4.3D^

In this set of experiments, we demonstrate that the phenotype caused by the directed expression of *Drp1* can be counteracted by knockdown via *Drp1-RNA*, and the phenotype caused by knockdown via *Drp1-RNA* can be counteracted by the directed expression of *Drp1*. Both are compared to *lacZ* controls. The control *Ddc-Gal4^4.3D^*; *UAS-lacZ* critical class males were determined to have a median lifespan of 62 days (n = 308). The directed knockdown of *Drp1* by the *Ddc-Gal4* transgene results in a greater median lifespan of 70 days in 321 flies compared to the control as determined by log-rank (Mantel–Cox) test with *p*-value < 0.0001 (Figure 3). In contrast, the directed expression of *Drp1* by the *Ddc-Gal4* transgene results in a reduced median lifespan of 56 days in 255 flies compared to the control as determined by log-rank (Mantel–Cox) test with *p*-value < 0.0001 (Figure 3A). Furthermore, the *Ddc-Gal4 UAS-Drp1-RNAi UAS-lacZ* critical class males have a median lifespan of 70 days in 310 flies. The directed expression of *Drp1* along with *UAS-Drp1-RNAi* under the direction of the *Ddc-Gal4* transgene (*Ddc-Gal4*; *UAS-Drp1-RNAi*; *UAS-Drp1*) has a median lifespan of 64 days, similar to control (*Ddc-Gal4*; *UAS-lacZ*) with a *p* value of 0.0633 as determined by the Log-rank Mantel–Cox test with a Bonferroni correction (Figure 3A). The *Ddc-Gal4 UAS-Drp1 UAS-lacZ* critical class males have a median lifespan of 58 days in 294 flies. The inhibition of *Drp1* along with *UAS-Drp1* under the direction of the *Ddc-Gal4* transgene (*Ddc-Gal4*; *UAS-Drp1*; *UAS-Drp1-RNAi*) has a median lifespan of 64 days (n = 327), similar to control (*Ddc-Gal4*; *UAS-lacZ*) with a *p* value of 0.0582 as determined by the Log-rank Mantel–Cox test with a Bonferroni correction (Figure 3C). The non-linear fitting of the climbing ability curve shows the *Drp1* expression and inhibition both have compromised the climbing ability phenotype compared to control at 95% CI (*p* < 0.0001) (Figure 3B,D). The climbing ability curve of *Ddc-Gal4 UAS-Drp1-RNAi UAS-Drp1* and *Ddc-Gal4 UAS-Drp1 UAS-Drp1-RNAi* is very close to the control (*Ddc-Gal4*; *UAS-lacZ*) as determined by the non-linear fitting of the climbing curve at a 95% CI at *p*-value 0.2752 and 0.0589, respectively. The co-expression of *Drp1-RNAi* along with *Drp1* ectopic expression in flies has resulted in phenotypes that are similar to the control and suggests that these phenotypes are primarily due to the changes in the expression of *Drp1*.

In Figure 3A,B, the *Ddc-Gal4* and the *Drp1-RNAi* genes were maternally contributed. When compared with *lacZ*, *Drp1* acted to reduce the enhanced median lifespan and reduced climbing of *Drp1-RNAi* knockdown. In Figure 3C,D, the *Ddc-Gal4* and the *Drp1* genes were maternally contributed. When compared with *lacZ*, *Drp1-RNAi* knockdown resulted in the rescue of the climbing defect, the *Ddc-Gal4 UAS-lacZ* control, with little change to lifespan.

### 3.4. Alteration of the Expression of Buffy and Debcl in Combination with Drp1 Directed by the Ddc-Gal4^4.3D^ Transgene

The control *Ddc-Gal4^4.3D^*; *UAS-Drp1*; *UAS-lacZ* critical class males were determined to have a median lifespan of 58 days (n = 282). The overexpression of *Buffy* along with *UAS-Drp1* under the direction of the *Ddc-Gal4* transgene (*Ddc-Gal4*; *UAS-Drp1*; *UAS-Buffy*) has a median lifespan of 68 days (n = 255), much higher compared to control with a *p* value of <0.0001 as determined by log-rank (Mantel–Cox) test with a Bonferroni correction. The knockdown of *Buffy* along with *UAS-Drp1* under the direction of the *Ddc-Gal4* transgene (*Ddc-Gal4*; *UAS-Drp1*; *UAS-Buffy-RNAi*) has a median lifespan of 52 days (n = 274), much less compared to the control (Figure 4A) with a *p* value 0.0125 as determined by log-rank Mantel–Cox test with a Bonferroni correction. The overexpression of *Buffy* in neurons rescued the early onset of impairment in the climbing ability of *Ddc-Gal4*; *UAS-Drp1* flies. The non-linear fitting of the climbing curve shows *Buffy* overexpression has rescued the climbing ability defect compared to control at 95% CI (*p* < 0.0001) (Figure 4B). The knockdown of *Buffy* by *Ddc-Gal4 UAS-Drp1*; *UAS-Buffy-RNAi* further contributes to loss of the climbing ability throughout the life of critical class flies compared to control at 95% CI at a *p*-value 0.0125 (n = 50) (Figure 4B).

The overexpression of *Debcl* along with *UAS-Drp1* under the direction of *Ddc-Gal4* transgene (*Ddc-Gal4*; *UAS-Drp1*; *UAS-Debcl^EY05743^*) has a median lifespan of 60 days (n = 331 flies), similar to the control (Figure 4A) with a *p* value at 0.0114 as determined by log-rank Mantel–Cox test with Bonferroni correction. The inhibition of *Debcl*, along with *UAS-Drp1* under the direction of the *Ddc-Gal4* transgene (*Ddc-Gal4*; *UAS-Drp1*; *UAS-Debcl-RNAi^v47515^*) has a median lifespan of 66 days (n = 303), much higher compared to the control (Figure 4A) with a *p* value at 0.0004 as determined by log-rank Mantel–Cox test with Bonferroni correction. The non-linear fitting of the climbing curve shows that *Debcl* overexpression has no change in the climbing ability defect compared to control at 95% CI (*p* = 0.3293) (Figure 4B). The knockdown of *Debcl* by *Ddc-Gal4 UAS-Drp1*; *UAS-Debcl-RNAi^v47515^* has rescued the climbing ability throughout the life of critical class flies compared to control at 95% CI at a *p*-value 0.0057 (n = 50) (Figure 4B).

### 3.5. Altering the Expression of Buffy and Debcl along with Drp1-RNAi by Ddc-Gal4^4.3D^ Transgene

The control *Ddc-Gal4^4.3D^*; *UAS-Drp1-RNAi*; *UAS-lacZ* critical class males were determined to have a median lifespan of 70 days (n = 323). The overexpression of *Buffy* along with *UAS-Drp1-RNAi* under the direction of the *Ddc-Gal4* transgene (*Ddc-Gal4*; *UAS-Drp1-RNAi*; *UAS-Buffy*) has a median lifespan of 64 days (n = 308), much lower compared to control with a *p* value of <0.0001 as determined by log-rank (Mantel–Cox) test with a Bonferroni correction. The co-inhibition of *Buffy* and *Drp1* under the direction of the *Ddc-Gal4* transgene (*Ddc-Gal4*; *UAS-Drp1-RNAi*; *UAS-Buffy-RNAi*) has a median lifespan of 62 days (n = 273), much less compared to the control (Figure 5A) with a *p* value at <0.0001 as determined by log-rank Mantel–Cox test with a Bonferroni correction. The non-linear fitting of the climbing curve shows *Buffy* overexpression has rescued the climbing ability defect compared to control at 95% CI (*p* < 0.0001) (Figure 5B). The inhibition of *Buffy* by *Ddc-Gal4 UAS-Drp1-RNAi*; *UAS-Buffy-RNAi* further contributes to loss of the climbing ability throughout the life of critical class flies compared to control at 95% CI at a *p*-value < 0.0001 (n = 50) (Figure 5B).

The overexpression of *Debcl* along with *UAS-Drp1-RNAi* under the direction of *Ddc-Gal4* transgene *(Ddc-Gal4*; *UAS-Drp1-RNAi*; *UAS-Debc^EY05743^*) has a median lifespan of 68 days (n = 156 flies), similar to the control (Figure 5A) with a *p* value at 0.0003 as determined by log-rank Mantel–Cox test with Bonferroni correction. The inhibition of *Debcl,* along with *UAS-Drp1-RNAi* under the direction of the *Ddc-Gal4* transgene (*Ddc-Gal4*; *UAS-Drp1-RNAi*; *UAS-Debcl-RNA^v47515^)* results in a median lifespan of 72 days (n = 321), higher compared to the control (Figure 5A) with a *p* value at 0.0211 as determined by log-rank Mantel–Cox test with Bonferroni correction. The non-linear fitting of the climbing curve shows that *Debcl* overexpression has further increased the climbing ability defect compared to control at 95% CI (*p* = 0.0004) (Figure 5B). The inhibition of *Debcl* by *Ddc-Gal4 UAS-Drp1-RNAi*; *UAS-Debcl-RNAi^v47515^* has rescued the climbing ability throughout the life of critical class flies compared to control at 95% CI at a *p*-value 0.0211 (n = 50) (Figure 5B).

## 4. Discussion

The protein product of the *Drp1* gene, along with the participation of other mitochondrial protection proteins, is involved in the processes of mitochondrial fission, apoptosis, and mitophagy. Excessive mitochondrial fragmentation can be associated with dysfunctional metabolic diseases, whereas a “hyper-fused” mitochondrial network can serve to protect cells from metabolic insult and autophagy [27]. In the skeletal muscle of mice, *Drp1* overexpression can cause a severe impairment of post-natal muscle growth as the production of protein may become attenuated and growth hormone pathways may be downregulated [28]. Conditions of high fat and/or high glucose levels can cause excessive oxidative stress along with mitochondrial fragmentation as mediated by the Drp1 protein [29,30]. These phenotypes are similar to the increased activity of *Drp1* as observed with Cos and PC12 cells [31]. In humans, protein kinase A (PKA) can phosphorylate and inactivate the pro-apoptotic Bcl-2 family member protein Bad [32] and the Drp1 protein [33] in a complex effort to promote cell survival. The effect of *Drp1* overexpression and, consequently, excessive mitochondrial fragmentation can be toxic to many physiological processes.

The Bcl-2 family proteins assist the pro-fission activity of the Drp1 protein during apoptosis in nematodes and mammals [34]. However, in non-apoptotic cells of mammals, the Bcl-2 family proteins have both pro-fission and pro-fusion activities. The overexpression of *Drp1* in selected neurons, along with the overexpression of *Buffy* or the inhibition of *Debcl*, has resulted in an increase in median lifespan and the ability to climb over the increased lifespan. In complementary experiments, *Buffy* knockdown and *Debcl* overexpression have resulted in reduced lifespans accompanied with impaired climbing abilities, consistent with the conclusion that *Buffy* can function as the antithesis of *Debcl* [12]. The rescue of the *Drp1* expression phenotype is in accordance with the role of the Buffy protein as “the guardian of mitochondria”. As proteins, Buffy can interact with Debcl to inhibit *Debcl*-induced cell death. As a mechanism, this process could be due to decreased activity of the Debcl protein, which influences cooperation with the Drp1 protein in the promotion of cell death [13]. The pro-apoptotic Debcl protein acts to induce apoptosis through a caspase-independent mechanism that triggers the release of Cytochrome C in an activity that resembles the loss of *Drp1* [6]. The overexpression of *Debcl* and *Drp1* together in selected neurons does not alter the phenotype generated by the overexpression of *Drp1* without *Debcl*. This may not be surprising as Drp1 and Debcl proteins seem to cooperate to promote apoptosis. Indeed, this and earlier studies have demonstrated that *Drp1* can play various roles in mitochondrial fragmentation and apoptosis to act in concert with anti- and pro-survival proteins, dependent upon the stimuli.

The directed knockdown in *Drosophila melanogaster* of *Drp1*, via *Drp1-RNAi*, in a subset of neurons results in an accelerated age-dependent loss in climbing ability, a phenotype strongly associated with the modelling of age-related disease in flies. The overexpression of *Buffy* in neurons that co-express *Drp1-RNAi* led to a decrease in the median lifespan accompanied with, or perhaps balanced by, a rescue of the impaired locomotor ability. The recovery in age-dependent climbing ability over time may be evidence of a complicated regulatory relationship. A study shows *Drp1* inhibition reduces the total accumulation of pro-apoptotic Bcl-2 protein, Bax, on the mitochondria outer membrane in HeLa cell lines [8]. This intermediate phenotype was not expected but may be important in the determination of the pathology of neurological diseases. The knockdown of anti-apoptotic *Buffy* or overexpression of pro-apoptotic *Debcl* enhanced the phenotype induced through the knockdown of *Drp1* in *Ddc-Gal4* expressing neurons. The interaction of *Bax* with *Drp1* in mammals seems to be conserved with this relationship. Drp1 protein interacts directly and indirectly with the Bcl-2 family protein to facilitate the permeability of the mitochondrial outer membrane in apoptotic cells [34]. Overall, we believe that we have established that *Buffy* confers a survival advantage to flies overexpressing *Drp1* and provides a partially rescued intermediate phenotype in flies with a knockdown of *Drp1* function.

## 5. Conclusions

Closely associated with cell death pathways in neurons, *Drp1* has recently been associated with ALS, Parkinson’s disease, and age-related disease. Our studies demonstrate that the directed overexpression and knockdown of *Drp1* activity in selected neurons can phenocopy some neurodegenerative-like symptoms in Drosophila and, therefore, may represent a novel model of disease. Importantly, the decrease in lifespan and age-dependent loss in climbing ability observed with overexpression of *Drp1* in flies is “rescued” to near controls either by overexpression of *Buffy* or by *Debcl* knockdown. The age-dependent loss of climbing ability in flies expressing *Drp1-RNAi* can be rescued by *Buffy* overexpression or *Debcl-RNAi*-directed knockdown. Future studies of these interactions will be required to chart out a pathway for *Drp1* and interactions with *Buffy* and *Debcl* in Drosophila and, importantly, the molecular changes associated with the loss-of-function of these genes in the development, function, and aging of the organism.

## Figures and Tables

**Figure 1 biomolecules-14-01089-f001:**
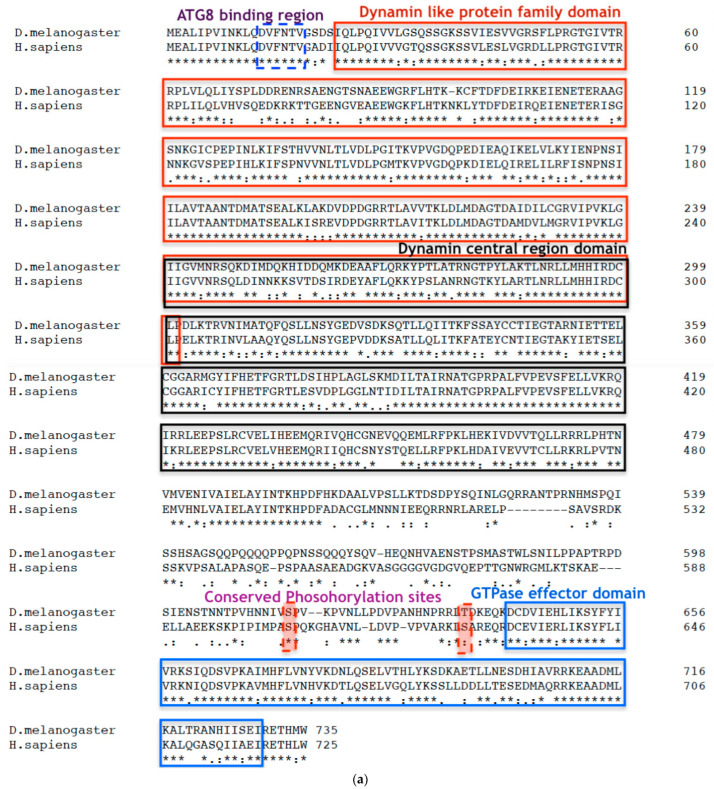
*Drp1* is evolutionarily conserved between Drosophila and humans. (**a**) Clustal Omega multiple sequence alignment of *D. melanogaster* Drp1 (NP_608694.2) protein with the *H. sapiens* (NP_001265392.1) shows evolutionarily conserved domains identified using the NCBI Conserved Domain Database (CDD) and is further confirmed by the Eukaryotic Linear Motif (ELM) resource. The two well-documented phosphorylation sites are identified: S606 and S627 in dynamin-1-like protein (DLP-1) isoform 4 of *H. sapiens* and S616 and T637 in Drp1 of *D. melanogaster*. The asterisks indicate the residues that are identical, the colons indicate the conserved substitutions, and the dots indicates the semi-conserved substitutions. Colour differences indicate the chemical nature of amino acids: red indicates small hydrophobic (includes aromatic) residues; blue indicates acidic; magenta indicates basic; and green indicates basic with hydroxyl or amine groups. (**bi**) The original Dynamin-1-like protein (DLP-1) structure of H. sapiens (NP_001265392.1) from the NCBI structure database. (**bii**) The Phyre2 web portal for protein modelling, prediction, and analysis mediated the development of a model of the Drp1 protein of *D. melanogaster* (NP_608694.2) from a 76% identical protein with a confidence of 100%. The N terminus is coloured in Magenta; C terminus is coloured in Red, and a consensus ATG8 binding region at N terminus is coloured in orange.

**Figure 2 biomolecules-14-01089-f002:**
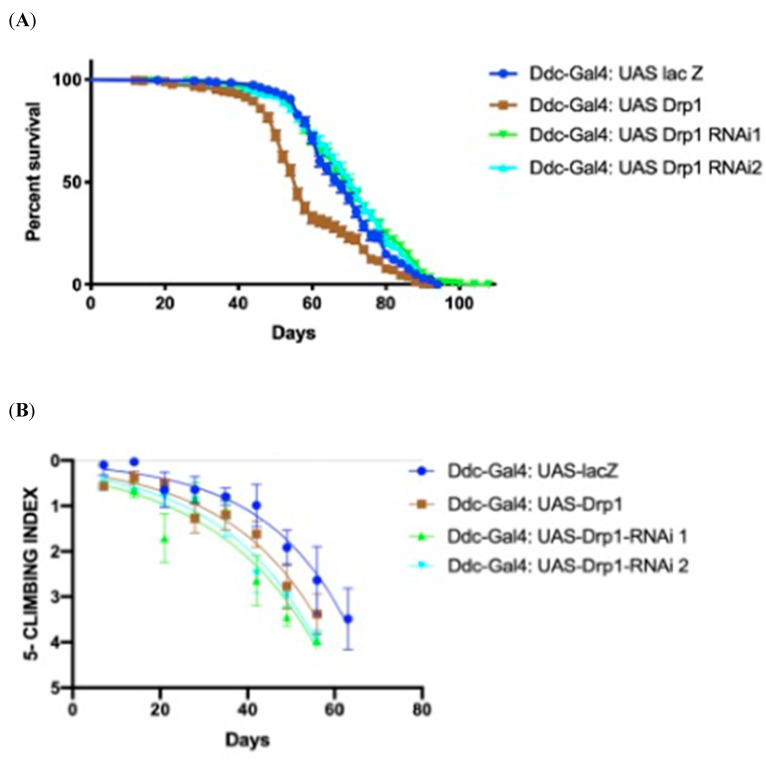
Altered *Drp1* expression under the control of *Ddc-Gal4^4.3D^* influences the survival and climbing ability of flies. (**A**) The GraphPad prism8 generated graph of the longevity assay for the expression of *Drp1*, *Drp1-RNAis* under the control of *Ddc-Gal4* transgene. The directed expression results in decreased median lifespan of 56 days compared to 68 days of control, as calculated by Log-rank Mantel–Cox test, with Bonferroni correction. The knockdown of *Drp1* under the control of the *Ddc-Gal4* transgene results in lifespan of 70 days with *UAS-Drp1-RNAi1* and 72 days with *UAS-Drp-RNAi2* compared to 68 days of control performed by Log-rank Mantel–Cox test, with Bonferroni correction. (**B**) The GraphPad prism8 generated graph of the climbing abilities of flies with overexpression of *Drp1*, the *Drp1-RNAis*, and the *lacZ* control. The climbing ability of *Drp1* overexpression and *Drp1*-*RNAi*s flies have decreased compared to control as determined in non-linear fitting of the climbing curve by 95% confidence interval (*p*-value < 0.0001).

**Figure 3 biomolecules-14-01089-f003:**
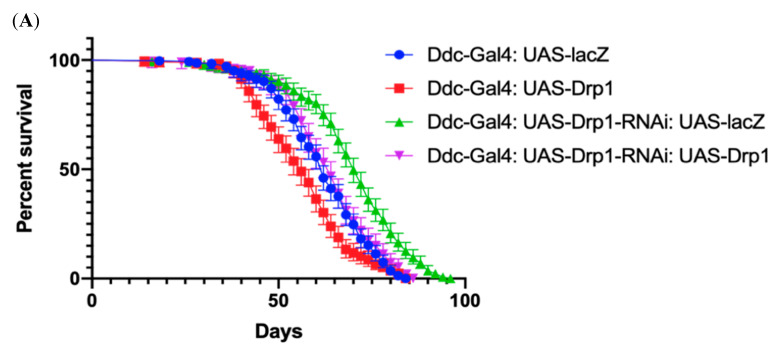
The ectopic expression of *Drp1-RNAi*, directed by *Ddc-Gal4^4.^*^3D^, can increase median lifespan and decrease climbing. (**A**) In control, *Ddc-Gal4^4.3D^ UAS-lacZ* critical class males resulted in a median life span of 62 days (n = 308). Expression of *Drp1* in *Ddc-Gal4^4.3D^* resulted in a median life span of 56 days (n = 310), much lower than the *lacZ*-expressing control; expression of *Drp1* in *Ddc-Gal4^4.3D^ UAS-Drp1-RNAi* resulted in a median life span of 64 days (n = 250) very similar to control (*Ddc/lacZ*) as determined by the Log-rank Mantel–Cox test (*p* value = 0.0633) with Bonferroni correction. The graph of the longevity assay was generated by GraphPad prism8. (**B**) The *Ddc-Gal4^4.3D^* flies express *UAS-lacZ* in control flies. The climbing abilities of *Ddc-Gal4^4.3D^ UAS-Drp1* expressing flies have decreased compared to control as determined in the non-linear fitting of the climbing curve by a 95% confidence interval (*p* < 0.0001). The flies’ climbing ability expressing *Drp1* in *Ddc-Gal4^4.3D^ UAS-Drp1-RNAi* transgene is similar to control as determined in the non-linear fitting of the climbing curve by a 95% confidence interval at *p* value = 1.309. The graph of longevity assay was generated by GraphPad prism8 non-linear regression curve. (**C**) In control, *Ddc-Gal4^4.3D^ UAS-lacZ* critical class males resulted in a median life span of 62 days (n = 308). Knockdown of *Drp1-RNAi* in *Ddc-Gal4^4.3D^* resulted in a median life span of 70 days (n = 321), much higher compared to the control; knockdown of *Drp1-RNAi* in *Ddc-Gal4^4.3D^ UAS-Drp1* resulted in a median life span of 64 days (n = 327), very similar to control (*Ddc/lacZ*) as determined by the Log-rank Mantel–Cox test (*p* value = 0.0582) with Bonferroni correction. The graph of the longevity assay was generated by GraphPad prism8. (**D**) The *Ddc-Gal4^4.3D^* flies express *UAS-lacZ* in control flies. The climbing abilities of *Ddc-Gal4^4.3D^ UAS-Drp1-RNAi* expressing flies have decreased compared to control as determined in the non-linear fitting of the climbing curve by a 95% confidence interval (*p* < 0.0001). The flies’ climbing ability expressing *Drp1-RNAi* in *Ddc-Gal4^4.3D^ UAS-Drp1* transgene is similar to control (*Ddc/lacZ*) as determined in the non-linear fitting of the climbing curve by a 95% confidence interval at *p* value = 0.0027. The graph of longevity assay was generated by GraphPad prism8 non-linear regression curve.

**Figure 4 biomolecules-14-01089-f004:**
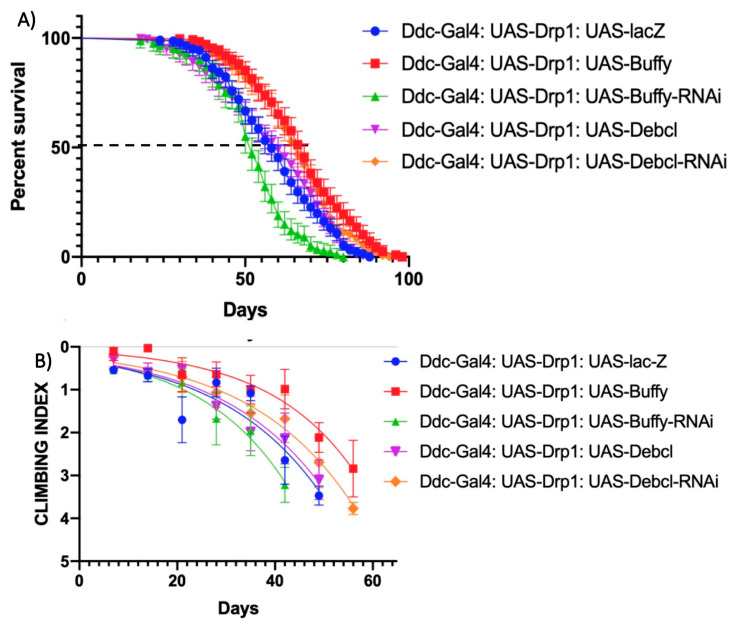
Altered expression of *Buffy* and *Debcl* can enhance and suppress climbing ability in *Drp1* over-expression flies. (**A**) In control, *Ddc-Gal4^4.3D^*; *UAS-Drp1 UAS-lacZ* critical class males resulted in a median life span of 58 days (n = 282). The overexpression of *Buffy* results in a median lifespan of 68 days (n = 375) compares to 58 days of control (*p* value = 0.0002); the knockdown of *Buffy* directed by the *Ddc-Gal4^4.3D^ UAS-Drp1* transgene results in the median lifespan of 52 (n = 274), much less compared to control, determined by Log-rank Mantel–Cox test at *p*-value < 0.0001, with Bonferroni correction. The overexpression of *Debcl^EY05743^* results in a median lifespan of 60 days (n = 331) similar to 58 days of control determined by Log-rank Mantel–Cox test at *p*-value 0.3293; the inhibition of *Debcl* directed by the *Ddc-Gal4 UAS-Drp1* transgene result in the median lifespan of 66 (n = 303); much higher than control, determined by Log-rank Mantel–Cox test at *p* value 0.0057, with Bonferroni correction. (**B**) The GraphPad prism8 generated graph of the climbing abilities of *Ddc-Gal4 UAS-Drp1* flies with the expression of *Buffy*, *Buffy-RNAi*, *Debcl^EY05743^*, *Debcl-RNAi^v47515^* and control. The climbing abilities of flies overexpressing *Buffy* have rescued compared to control as determined in the climbing curve’s non-linear fitting by a 95% confidence interval (*p* < 0.0001). The climbing ability of the flies was further weakened by the knockdown of *UAS-Buffy-RNAi* as determined in the non-linear fitting of the climbing curve by a 95% confidence interval at a *p*-value 0.0125 and 0.03293, respectively (n = 50). The climbing abilities of flies expressing *Debcl-RNAi^v47515^* has been rescued compared to control as determined by the non-linear fitting of the climbing curve by a 95% confidence interval (*p* value = 0.0057). The graph of longevity assay was generated by GraphPad prism8 non-linear regression curve.

**Figure 5 biomolecules-14-01089-f005:**
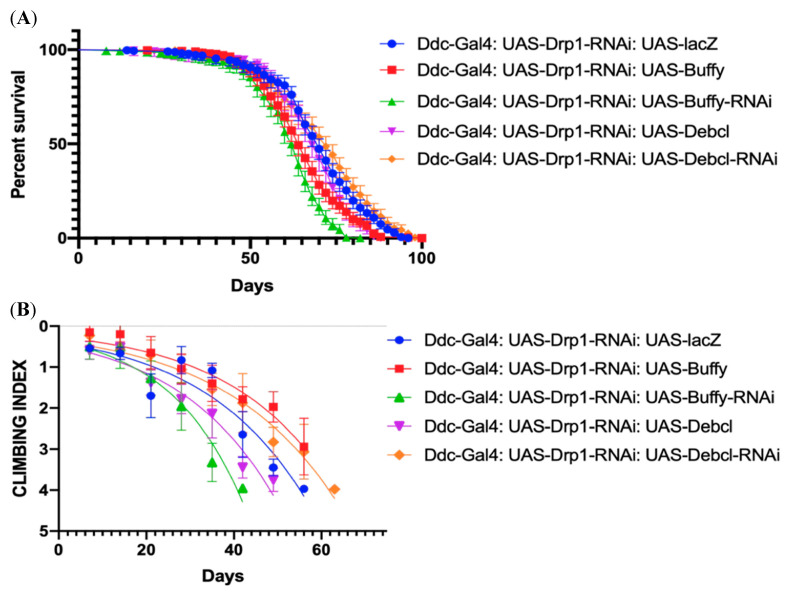
Altered expression of *Buffy* and *Debcl* can enhance and suppress climbing ability in *Drp1* knockdown flies. (**A**) In control, *Ddc-Gal4^4.3D^*; *UAS-Drp1* transgene results in the median lifespan of 62 (n = 273), determined by Log-rank Mantel–Cox test at *p*-value < 0.0001, with Bonferroni correction. The overexpression of *Debcl^EY05743^* results in a median lifespan of 68 days (n = 331), much higher compared to control as determined by Log-rank Mantel–Cox test at *p*-value 0.0003; the inhibition of *Debcl* directed by the *Ddc-Gal4*; *UAS-Drp1* transgene result in the median lifespan of 72 (n = 303); similar to control, determined by Log-rank Mantel–Cox test at *p*-value 0.021, with Bonferroni correction. (**B**) The GraphPad prism8 generated graph of the climbing abilities of *Ddc-Gal4 UAS-Drp1* flies with the expression of *Buffy, Buffy-RNAi*, *Debcl^EY05743^**, Debcl-RNAi^v47515^* and control. The climbing abilities of flies overexpressing *Buffy* have rescued compared to control as determined in the climbing curve’s non-linear fitting by a 95% confidence interval (*p* < 0.0001). The climbing ability of the flies has further diminished through the ectopic expression of *UAS-Buffy-RNAi* and *UAS-Debcl^EY05743^* as determined in the non-linear fitting of the climbing curve by a 95% confidence interval at a *p*-value 0.0004 and 0.0002 respectively (n = 50). The climbing abilities of flies expressing *Debcl-RNAi^v47515^* has been rescued compared to control as determined by the non-linear fitting of the climbing curve by a 95% confidence interval (*p* value < 0.0001). The graph of longevity assay was generated by GraphPad prism8 non-linear regression curve.

## Data Availability

Data are contained within the article.

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
