# Peer review of "Bcl-2 Orthologues, Buffy and Debcl, Can Suppress Drp1-Dependent Age-Related Phenotypes in Drosophila"

_biomolecules, 2024, doi:10.3390/biom14091089_

Round 1

Reviewer 1 Report

Comments and Suggestions for Authors

A very interesting study was carried out on a current topic. However, the presentation of the results and the design of the article need significant improvement:

1. Check the names of genes throughout the whole article, so that the genes are written in cursive, while proteins and regular text is not (ex. lines 15, 285 etc.)

2. Check English (ex. lines 46, 57, 67, 76 etc.)

3. Rewrite the sentence on line 12 to make it less repetitive, consider “overexpression and inhibition of Drp1”; overall the Abstract section seems repetitive

4. Latest reference is from 2020, but a lot of research has been done since then; for ex: lines 52-53 - in recent years, many works have been published on these topics.

5. Figure numbers don’t match (Figure 1 is written twice)

6. All abbreviations must be explained (ex. CI, JNK, etc.); all explanations of abbreviations must be made (or transferred) at the first mention (ex. LIR from the line 162 to the line 109

7. In line 69 translocation of Drp1 is mentioned, but the reason for the translocation is not explained, it might be helpful to give more details

8. In line 74 adjustment of Drp1 is viewed as essential for development of therapies, and in line 78 alteration of Drp1 expression leads to degenerative phenotypes. While being true as separate statements, writing it like this lacks logic

9. ND-like phenotype on line 88 is meant to be PD-like?

10. Line 100: Add an abbreviation CDD

11. Rewrite the sentence on line 198 to show that altering the Drp1 expression in both directions compromises climbing abilities

12. Make clearer the difference between the experiments 2.2 and 2.3; and between experiments 2.4 and 2.5. Maybe it would be better to combine these pairs into two sections instead of four

13. Figure 6 shows the wrong data in 6B

14. The written explanation of the results of the Parkin experiment is not clear, therefore the discussion of the results also seems vague based on the results

15. It might be beneficial to change the title of the article, exclude mentioning parkin, but focus on Buffy and Debcl Bcl-2 homologues; the expression “suppresses...loss” in the context of the title is not very good

Reviewer 2 Report

Comments and Suggestions for Authors

Review reports biomolecules-3112243

A brief summary

The authors claim that Buffy's pro-survival effect can rescue Drp1-induced apoptosis. However, since the biochemistry data are lacking, the author should consider rewording the conclusion of the apoptosis pathway, which can be supported by bioinformatics data to connect the gene to the functional pathways.

General concept comments

1.     The sequence alignment and protein structure in Figure 1 come from publicly accessible data and should be presented in the supplemental materials or removed from the main figures.

2.     It is crucial to provide data on the knockdown efficiency validation of all RNAi lines. If these lines were used in another study, please cite the publications. If not, the authors should verify the knockdown using applicable methods such as RT-qPCR, western, or IF staining.

3.     For the overexpression lines involved in this study, it is essential to either cite references or perform experiments to determine protein levels. This will significantly enhance the robustness of the study.

4.     For the behavioral test, the group size is 50 for all the genotypes involved in this study. It’s suspected to have the same number from the start day to the end day, as shown on the x-axis of the climbing index plots. Given that controls and Drp1 overexpression flies die at day 50, the number should change along the progress rather than 50 flies throughout the test. If animal replacement happens, the author should indicate the replacement in the methods section. The protocol and explanation for this assay should be improved in the methods section.

Specific comments 

1.     The terminology about gain and loss function. In this study, knockdown should be the best term to describe the RNAi line, rather than the term loss of function or inhibition.

2.     There are two Figure 1.

3.     The climbing index's x-axis is “Days.” However, due to weekly behavioral tests, it’s better to have Weeks as the x-axis title.

4.     The lower part of the Figure is cropped.

5.     Based on the survival plot of the Ddc-Gal4:UAS LacZ group from Figure 1 (missing A) on page 8, Figure 2A, and Figure 3A, the survival curve cross with the x-axis is 95, 85, and 85. What does this mean?  Even the same control line will show different survival patterns; how can you explain all other data from this behavioral test?

6.     After Figure 3, no control line was included in the experiment. It’s better to have a health control to remind the reader of the effect of any gene knockdown or overexpression.

7.     Description of Drosophila stocks and media: Lines 115-121 are hard to read.

Round 2

Reviewer 2 Report

Comments and Suggestions for Authors

Your work, though currently limited to behavioral tests, holds significant potential and is a valuable addition to our scientific community. However, the absence of genotyping evidence for the transgene models and the lack of clarity on the gene overexpression or knockdown effects make the conclusion less convincing. 

If the heterozygous animals were used after breeding, then the genotyping should be done at whatever level. The authors need to provide genotyping evidence for their transgene models. I know the fly lines used in this study are already in stock and validated. However, the combinations are novel, and the genotypes need to be clarified.

For the double—or triple-transgene models, how does the gene overexpression or knockdown affect the expression of the other genes at the mRNA or protein level? 

Author Response

Responses to Reviewer Biomolecules Minor Revisions 2

Reply to the Review Report (Reviewer 1)

Comments 1: Your work, though currently limited to behavioral tests, holds significant potential and is a valuable addition to our scientific community (Response: Thank you). However, the absence of genotyping evidence for the transgene models and the lack of clarity on the gene overexpression or knockdown effects make the conclusion less convincing.

Response 1: To the section Drosophila stocks and media, the following was added:

In brief, a standard “double balancer chromosome” line of the genotype w1118; L/CyO; Ki/TM3 was used to generate intermediate lines  w1118; Ddc-Gal4/CyO; Ki/TM3; w1118; L/CyO; UAS-Drp1/TM3 and w1118; L/CyO; UAS-Drp1-RNAi/TM3 which were then used to generate the w1118; Ddc-Gal4/CyO; Ki/TM3; UAS-Drp1/TM3 and w1118; Ddc-Gal4/CyO; UAS-Drp1-RNAi/TM3 lines.

Comments 2:  If the heterozygous animals were used after breeding, then the genotyping should be done at whatever level. The authors need to provide genotyping evidence for their transgene models. I know the fly lines used in this study are already in stock and validated. However, the combinations are novel, and the genotypes need to be clarified.

Response 2: The careful use of balanced lines to generate the w1118; Ddc-Gal4/CyO; Ki/TM3; UAS-Drp1/TM3 and w1118; Ddc-Gal4/CyO; UAS-Drp1-RNAi/TM3 lines should clarify generation of the genotypes. We believe that the functional interaction between Drp1 and Drp1-RNAi as seen in Figure 3 is supportive.

Comments 3: For the double—or triple-transgene models, how does the gene overexpression or knockdown affect the expression of the other genes at the mRNA or protein level?

Response 3: We hope that those colleagues with sufficient resources, or perhaps ourselves, if funding improves, will be able to conduct such investigations in the future. Our plans are to continue to investigate the roles Drp1 during aging in combination with a number of strong candidate genes.
